# Associations between environmental factors and running performance: An observational study of the Berlin Marathon

Katja Weiss[1], David Valero[2], Elias Villiger[1], Volker Scheer[2], Mabliny Thuany[3], Felipe J. Aidar[4,5], Raphael Fabrício de Souza[4,5], Ivan Cuk[6], Pantelis T. Nikolaidis[7], Thomas Rosemann[1], Beat Knechtle[1,8]*

1 Institute of Primary Care, University of Zurich, Zurich, Switzerland, 2 Ultra Sports Science Foundation, Pierre-Benite, France, 3 Department of Physical Education, State University of Para, Pará, Brazil, 4 Group of Studies and Research of Performance, Sport, Health and Paralympic Sports—GEPEPS, The Federal University of Sergipe—UFS, São Cristovão, Sergipe, Brazil, 5 Department of Physical Education, Federal University of Sergipe—UFS, São Cristovão, Sergipe, Brazil, 6 Faculty of Sport and Physical Education, University of Belgrade, Belgrade, Serbia, 7 School of Health and Caring Sciences, University of West Attica, Athens, Greece, 8 Medbase St. Gallen Am Vadianplatz, St. Gallen, Switzerland

* beat.knechtle@hispeed.ch

**Data Availability Statement:** Weather data were from the 'Deutscher Wetterdienst' website (https://opendata.dwd.de/climate_environment/CDC/observations_germany/climate/hourly/) specifically

## Abstract

Extensive research has delved into the impact of environmental circumstances on the pacing and performance of professional marathon runners. However, the effects of environmental conditions on the pacing strategies employed by marathon participants in general remain relatively unexplored. This study aimed to examine the potential associations between various environmental factors, encompassing temperature, barometric pressure, humidity, precipitation, sunshine, cloud cover, wind speed, and dew point, and the pacing behavior of men and women. The retrospective analysis involved a comprehensive dataset comprising records from a total of 668,509 runners (520,521 men and 147,988 women) who participated in the 'Berlin Marathon' events between the years 1999 and 2019. Through correlations, Ordinary Least Squares (OLS) regression, and machine learning (ML) methods, we investigated the relationships between adjusted average temperature values, barometric pressure, humidity, precipitation, sunshine, cloud cover, wind speed, and dew point, and their impact on race times and paces. This analysis was conducted across distinct performance groups, segmented by 30-minute intervals, for race durations between 2 hours and 30 minutes to 6 hours. The results revealed a noteworthy negative correlation between rising temperatures and declining humidity throughout the day and the running speed of marathon participants in the 'Berlin Marathon.' This effect was more pronounced among men than women. The average pace for the full race showed positive correlations with temperature and minutes of sunshine for both men and women. However, it is important to note that the predictive capacity of our model, utilizing weather variables as predictors, was limited, accounting for only 10% of the variance in race pace. The susceptibility to temperature and humidity fluctuations exhibited a discernible increase as the marathon progressed. While weather conditions exerted discernible influences on running speeds and outcomes, they did not emerge as significant predictors of pacing.

for the 'Alexanderplatz' location, which is near the race venue. The athlete information was directly sourced from the official 'Berlin Marathon' website (www.bmw-berlin-marathon.com/). We acquired the complete dataset encompassing all races from 1999 to 2019 in JSON format, which was subsequently transformed into Excel format utilizing a tailored Python script. The datasets used and analyzed during the current study are available from Institute of Primary Care, University of Zurich, Zurich, Switzerland.

**Funding:** The author(s) received no specific funding for this work.

**Competing interests:** The authors have declared that no competing interests exist.

## Introduction

Marathons have been attracting a growing number of participants, and consequently, the number of athletes, including older ones, has been rising [1]. Several performance-related variables, including load control, have been explored to achieve better performance [2]. However, weather conditions strongly impact performance, particularly in long-distance races such as marathons [2, 3].

Weather variables such as temperature [3–5], precipitation [3, 6], wind direction and wind speed [3, 7], sunshine [6, 8], barometric pressure [3, 6, 8], dew point [4], cloud cover [8] and humidity [9] have been investigated. Among these variables, the ambient temperature significantly influences marathon running performance [3, 10]. Concerning the impact of temperatures on marathon running performance, it has been reported that differences between genders [6, 9, 11] and performance levels of athletes [3, 11, 12] seem to exist. The influence of temperature has been mainly reported for amateur marathoners [6, 13], where increasing temperatures negatively impacted slower runners more than faster runners [3, 9]. Elite marathoners, however, seemed to achieve the fastest race times on race days with higher-than-average temperatures at the 'Berlin Marathon' (mean average temperature 12.7˚C, mean maximum temperature 18.02˚C) [8]. Other environmental conditions also play significant roles in marathon performance. Precipitation can affect running conditions by making the course slippery and impacting the runner's ability to maintain a steady pace [3, 6, 8]. Wind direction and speed can either aid or hinder runners depending on whether they face headwinds or tailwinds during the race [3, 7]. Sunshine influences thermal stress and hydration needs [6, 8], while barometric pressure can impact oxygen availability, especially in events at varying altitudes [6, 8]. Dew point and humidity affect the body's ability to dissipate heat through sweat evaporation [4, 9], and cloud cover can mitigate the effects of direct sunlight, influencing the overall thermal load on the runners [8].

The 'Berlin Marathon' is the marathon race with the most world records in marathon running [14]. The effects of the environment on athletic performance have been reported for both elite and amateur athletes [6]. Thus, higher temperatures and longer sun exposure tended to slow down slower marathon runners more than elite marathon runners [6, 8]. Similarly, when analyzing performance in marathons and climatic factors, pacing in marathon running was highlighted as essential for a successful race outcome [15–17].

Pacing in marathon running has been investigated for different performance levels, such as elite [14, 18] and amateur [15, 19] marathoners. Generally, both elite and amateur marathoners showed a positive pacing profile with a decline in running speed during the race [17, 20] and the end spurt [14, 21, 22]. It has been reported that experience and training volume were predictive of smaller declines in running speed during the second half of a marathon [23].

Heat production via exercise is proportional to body mass [24]. In contrast, heat loss is related to the body's surface area [25]. Since women generally have a larger surface area-to-mass ratio than men, women should be able to disperse a higher percentage of excess heat produced during running. A study from the 'Chicago Marathon' showed that women were better pacers than men in amateur marathons, depending on the temperature. In contrast, no difference between elite men and women was shown [11]. In addition, a comparison between the 'World Marathon Majors Series' over the last 13 editions showed differences in the winner's pacing behavior. The authors concluded that the course topography and environmental conditions could influence the results [26]. A recent study investigating the top 1,000 runners in 12 editions of the 'New York City Marathon' showed that faster runners suffered a higher performance decline than slower ones when thermal exposure was identical [27]. The authors concluded that future studies should investigate the relationship between thermal conditions and pacing.

Therefore, the aim of the present study was to investigate a potential association between environmental conditions such as temperature, barometric pressure, humidity, precipitation, sunshine, cloud cover, wind speed and dew point, and average race pace among the 668,509 marathoners competing in the 'Berlin Marathon' between 1999 and 2019. Based on existing knowledge, we hypothesized to find a relationship between air temperature and running speed or pace (i.e., with increasing temperature, running speed would decrease), that the increasing temperature during the race day would differently affect the pacing of women compared to men, (i.e., women would be less affected than men) and that weather factors will lend themselves as a significant predictor for pacing.

## Materials and methods

### Approach

To assess our hypothesis, we procured data from the official 'Berlin Marathon' website (www.bmw-berlin-marathon.com/) encompassing participants' first and last names, gender, age, calendar year, and running times. It is worth noting that split times were not documented in the race records before 1999, and the year 2020 did not host a race due to the COVID-19 pandemic (www.bmw-berlin-marathon.com/).

### The race

The Berlin Marathon occurs between the middle and concluding weeks of September in Berlin, with an elevation of 34 meters above sea level (www.bmw-berlin-marathon.com/). During this period, the city encounters an average temperature of approximately 15˚C and an average humidity of about 75% (www.weather-atlas.com/en/germany/berlin-weather-september). The race route forms a sizeable loop through the historic city of Berlin, culminating at the iconic 'Brandenburger Tor.' The course's maximum elevation is 50 meters, with a maximum gain of 20 meters (www.bmw-berlin-marathon.com/). In detail, it climbs ~20 m with a slight uphill from 20–25 km and a bigger incline from 25–27 km then a downhill from 27–32 km. The start of the race is organized into time blocks ranging from 09:15 a.m. to 10:30 a.m., depending on the best time the starter states upon registration (www.bmw-berlin-marathon.com/).

### Subjects

The athlete information was directly sourced from the official 'Berlin Marathon' website (www.bmw-berlin-marathon.com/). We acquired the complete dataset encompassing all races from 1999 to 2019 in JSON format, which was subsequently transformed into Excel format utilizing a tailored Python script.

### Weather data

We sourced all weather information for the race day from the 'Deutscher Wetterdienst' website (https://opendata.dwd.de/climate_environment/CDC/observations_germany/climate/hourly/) specifically for the 'Alexanderplatz' location, which is near the race venue. We accessed hourly data encompassing air temperature (˚C), relative humidity (%), sunshine (duration in minutes), precipitation (mm), barometric pressure (mbar), cloud cover (duration in minutes), wind speed (km/h), and dew point (˚C) commencing from 9:00 a.m. and spanning 8 consecutive hours. This duration accommodated all official finishers with race times of up to 8 hours. The data was provided with integer resolution (no decimal points). No information was available concerning any alterations, changes, or incidences of any type in the measuring instruments used during the analyzed timeframe.

## Data processing

This study utilized two distinct datasets. The initial dataset encompassed comprehensive information about the 'Berlin Marathon' participants from 1999 to 2019. This included partial and full race times, formatted as h:min:s, gender, age, and marathon year. Given the focus on assessing the impact of weather conditions on performance groups' pacing, age data were excluded from the analysis. Nonetheless, gender-based variations were also explored. The second dataset documented weather conditions throughout each marathon held between 1999 and 2019. This encompassed hourly temperature measurements, atmospheric pressure, humidity, precipitation, wind speed, dew point, sunshine minutes, and cloud cover percentages for each hourly interval. These datasets were processed using a Google Colab notebook, utilizing Python 3.9 for statistical analysis, and creating results tables and visual charts. Race records underwent scrutiny to ensure data accuracy and coherence, with erroneous or incomplete entries excluded from the analysis.

## Statistical analysis

The entire race record was classified into performance time groups based on individual runners' race times. These groups were delineated as follows: under 2 hours 30 minutes, 2 hours 31 minutes to 3 hours, 3 hours 1 minute to 3 hours 30 minutes, 3 hours 31 minutes to 4 hours, 4 hours 1 minute to 4 hours 30 minutes, 4 hours 31 minutes to 5 hours, 5 hours 1 minute to 5 hours 30 minutes, 5 hours 31 minutes to 6 hours, and over 6 hours [22]. Importantly, each runner's record was exclusively allocated to a single group.

Subsequently, every record was assigned adjusted average weather values corresponding to their race duration on the event day. This approach enhanced the precision of weather condition representation for each runner's race, leading to heightened resolution and accuracy. Additionally, utilizing the original data for split and total distance times, the average running speed (in km/h) and pace (in min/km) were calculated for each record. A comprehensive descriptive statistical analysis combined pace, race times, and weather data. This encompassed Pearson correlation analysis, calculation of statistical parameters for total and partial race times, and depiction of weather factors for each time group using boxplot charts. A multivariate Ordinary Least Squares (OLS) linear regression model was first built, and then a second, more sophisticated XG Boost regression model was implemented. The eight weather factors were used as predictors in all cases, and the race pace was the model output. Partial Dependence Plots (PDP) and Shape Additive Explanatory Values (SHAP) techniques were then used to interpret the XG Boost model predictions. Statistical values were presented in tables and charts detailing average (mean) and standard deviation (SD), as well as maximum (max) and minimum (min) values for each category. The 'N' column denoted the sample count within each specific category. Assurance of Gaussian (normal) distribution for race times and paces was validated through histogram and probability density distribution plots, accompanied by statistical significance tests and computation of p-values. All analyses were carried out using the Python programming language (Python Software Foundation, https://www.python.org/), within a Google Colab notebook (https://colab.research.google.com/), and the Statistical Software for the Social Sciences (IBM SPSS v26. Chicago, Ill, USA).

## Ethics approval

The institutional review board of St. Gallen, Switzerland (EKSG 01/06/2010) approved this study. Since the study focused on analyzing publicly accessible data, informed consent was exempt.

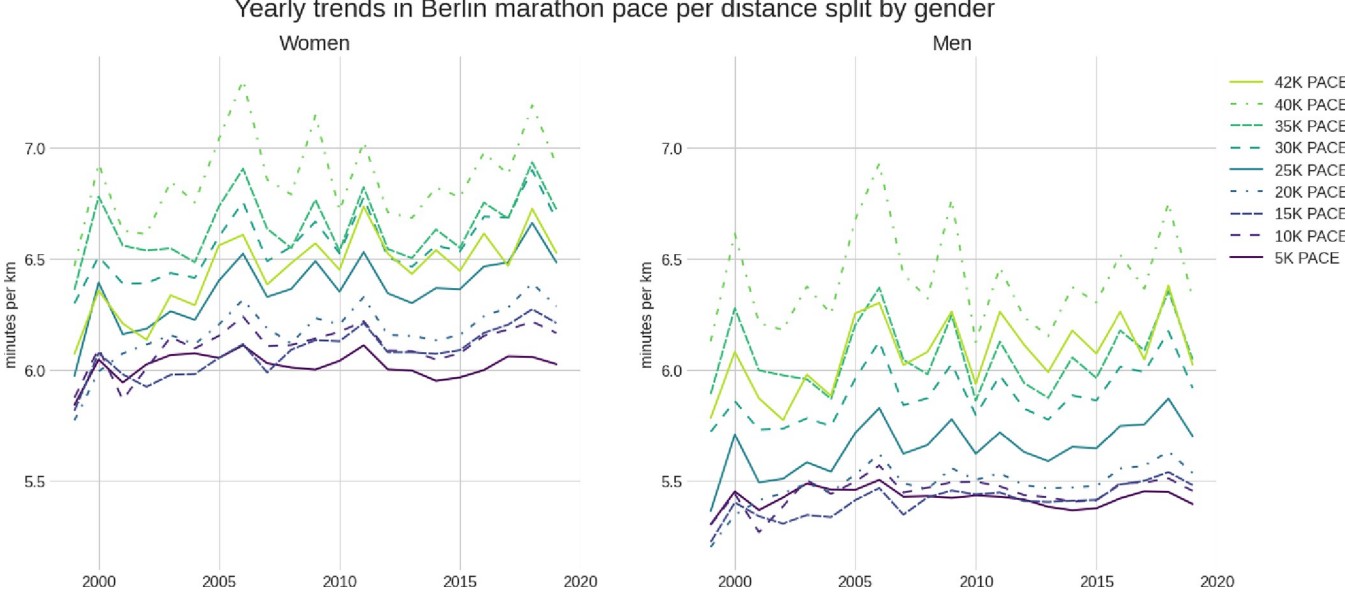

**Fig 1. Average marathon pace per distance split by gender between 1999 and 2019.**

## Results

A total of 668,509 runners' records (520,521 men and 148,988 women) from competitors between 1999 and 2019 were considered in this study. The mean running speed for all years and all runners was ~10.5 km/h, corresponding to a mean race time of ~4 hours at an average pace of 5 minutes and 53 seconds per km (5.89 min/km). **Fig 1** shows the average pace per split distance through the years for each gender. The mean running speed per split decreased with the increasing race distance but stayed almost constant through the first 15 or 20 km of the race, especially among men.

**Fig 2** shows the distributions of the average pace at each split and the full race by gender. The tails of the curves shift noticeably to the right from the 25 km split onwards.

**Table 1** summarizes the environmental conditions during the races between 1999 and 2019 where the mean temperature was at ~16°C.

**Fig 3** presents the trends of temperature, humidity, barometric pressure, wind speed, and dew point during the race hours. The temperature usually increased with the hour of the day, while humidity behaved oppositely.

**Fig 4** shows an equivalent boxplot with the weather conditions. The best runners experienced colder temperatures and higher humidity levels in the first three hours from 09:00 a.m. when the temperature was low and humidity high. Runners progressed slower and completed the race in the afternoon when the temperatures were higher and the humidity lower. The slowest groups (*i.e.*,6 h and 6 h and longer) see significant variations in the weather conditions throughout the 6 h, 7 h or up to 8 h, which is why their boxes are the largest in each plot. Regarding precipitation, the sample is too small (and the data has too little variance) to extract any insights.

**Fig 5** is a large correlation matrix of the split distance average paces including the full race distance *versus* the weather factors for both women and men. A positive correlation exists between all split average paces and the temperature (r = 0.12 to 0.22 for women, r = 0.18 to 0.29 for men). This correlation increases with the race distance, indicating a higher sensitivity of the pace to the temperature as the race evolves. This is more noticeable in men than in

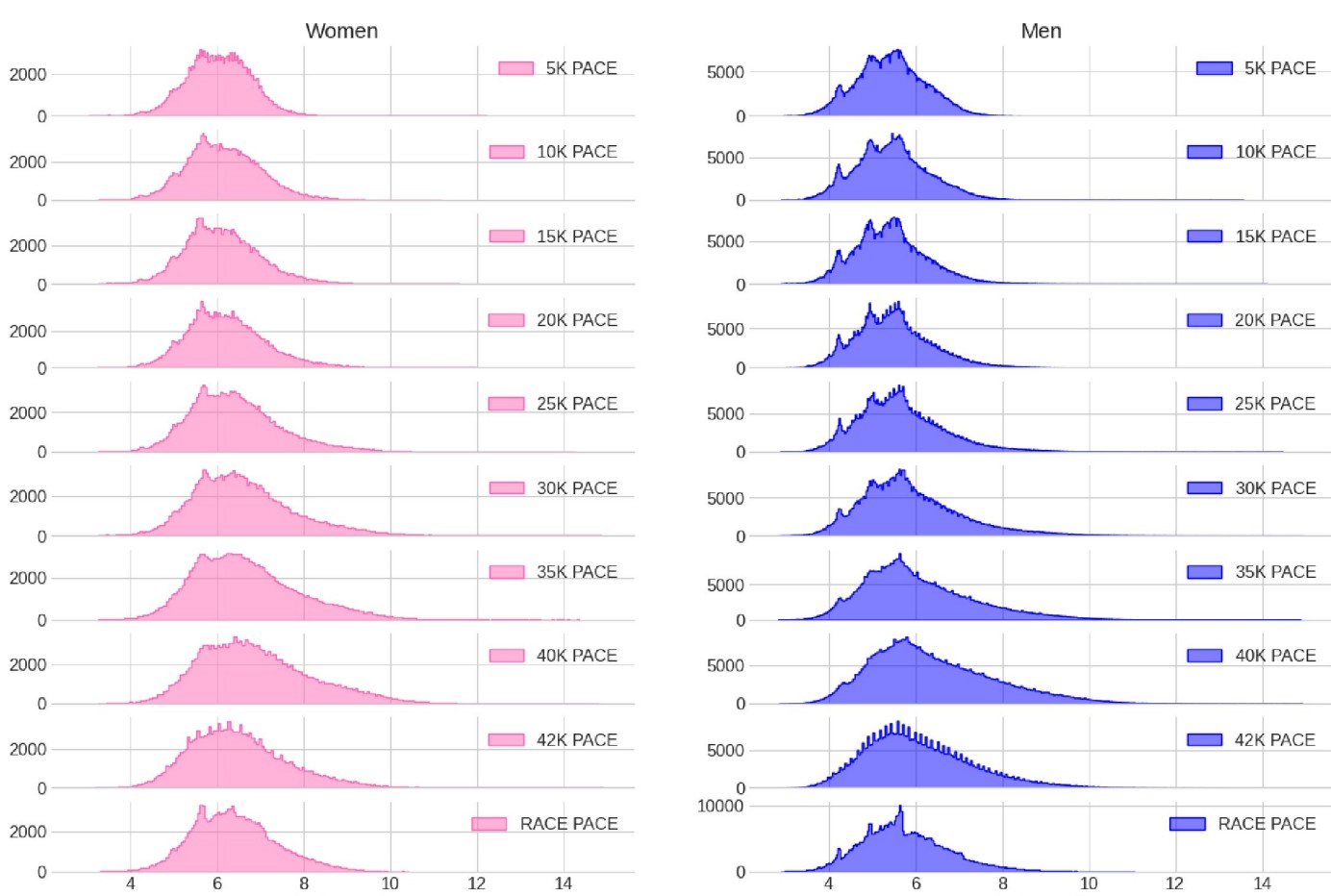

**Fig 2. Distributions of average pace per split and full race by gender in Berlin Marathon between 1999 and 2019.**

women. A negative correlation exists between all split average paces and the humidity (r = -0.08 to -0.15 for women, r = -0.14 to -0.22) with a similar (but opposite) behavior to temperature. These correlation coefficients are statistically significant (p<0.05). The main observation is that the average pace for the full race shows positive correlations with the temperature and the minutes of sunshine for both men and women. So these two factors slow down runners.

**Table 1. Basic details of weather-related factors during Berlin Marathons (1999–2019).**

|         | Temperature (˚C) | Pressure (mbar) | Humidity (%) | Precipitation (mm) | Sunshine (min) | Cloud cover (%) | Wind speed (km/h) | Dew Point (˚C) |
|---------|------------------|-----------------|--------------|--------------------|----------------|-----------------|-------------------|----------------|
| Mean    | 16               | 1015            | 69           | 0                  | 33             | 61              | 23                | 10             |
| SD      | 4                | 9               | 18           | 0                  | 27             | 37              | 10                | 3              |
| Minimum | 7                | 995             | 35           | 0                  | 0              | 0               | 6                 | 3              |
| 25%     | 13               | 1006            | 55           | 0                  | 0              | 25              | 17                | 9              |
| 50%     | 16               | 1018            | 70           | 0                  | 40             | 75              | 20                | 10             |
| 75%     | 18               | 1022            | 86           | 0                  | 60             | 100             | 28                | 12             |
| Maximum | 27               | 1032            | 97           | 2                  | 60             | 100             | 59                | 16             |

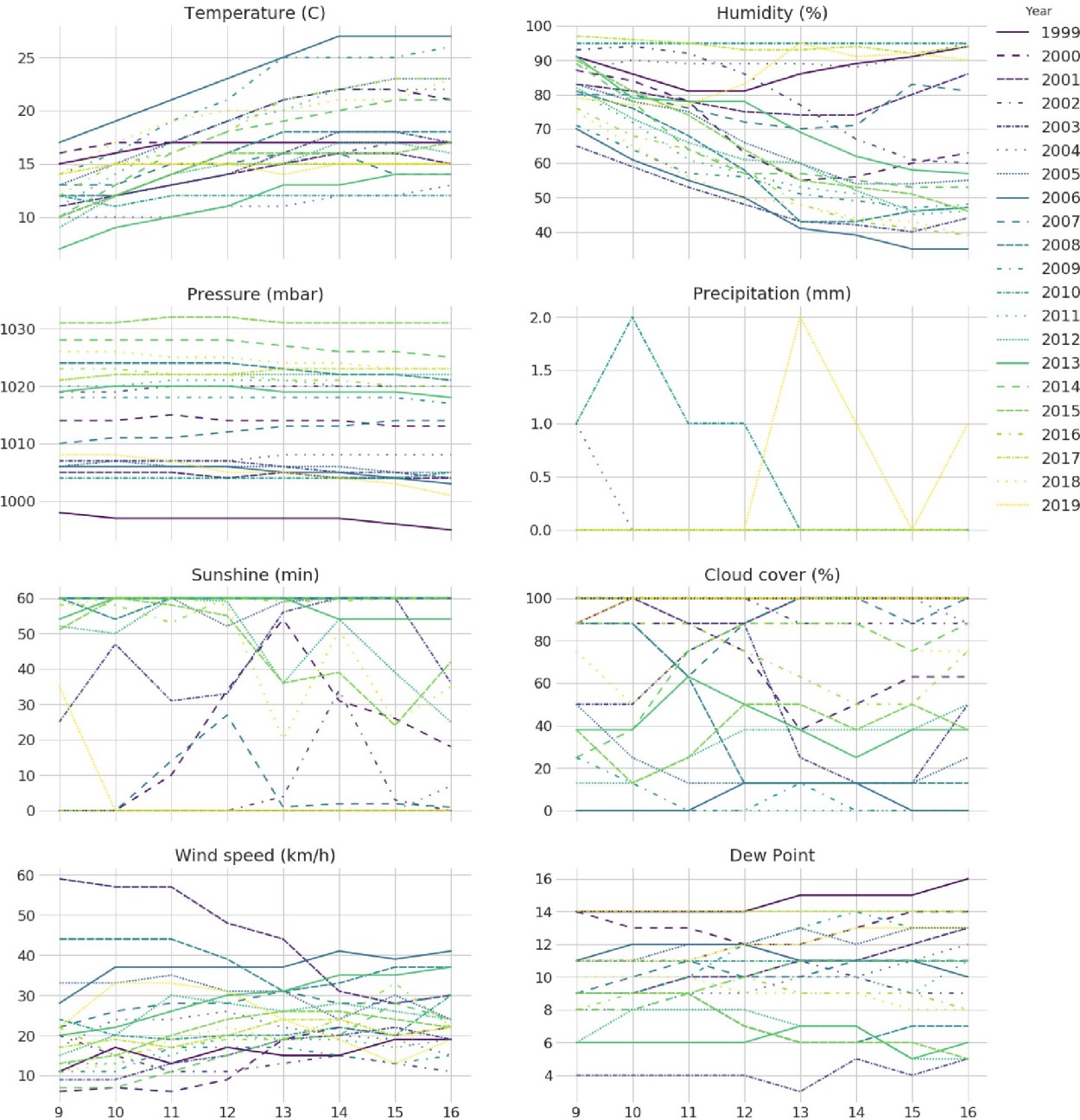

**Fig 3. Hourly weather conditions during Berlin Marathons between 1999 and 2019.**

The rest of the weather factors do not present a noticeable correlation with the average paces through the split distances.

We used a multivariate Ordinary Least Squares (OLS) regression to build a predictive model of the race average pace as a function of the weather factors. **Table 2** shows that the model with an R-squared of 0.169 can only predict 17% of the race pace with weather factors as predicting variables (Model 1).

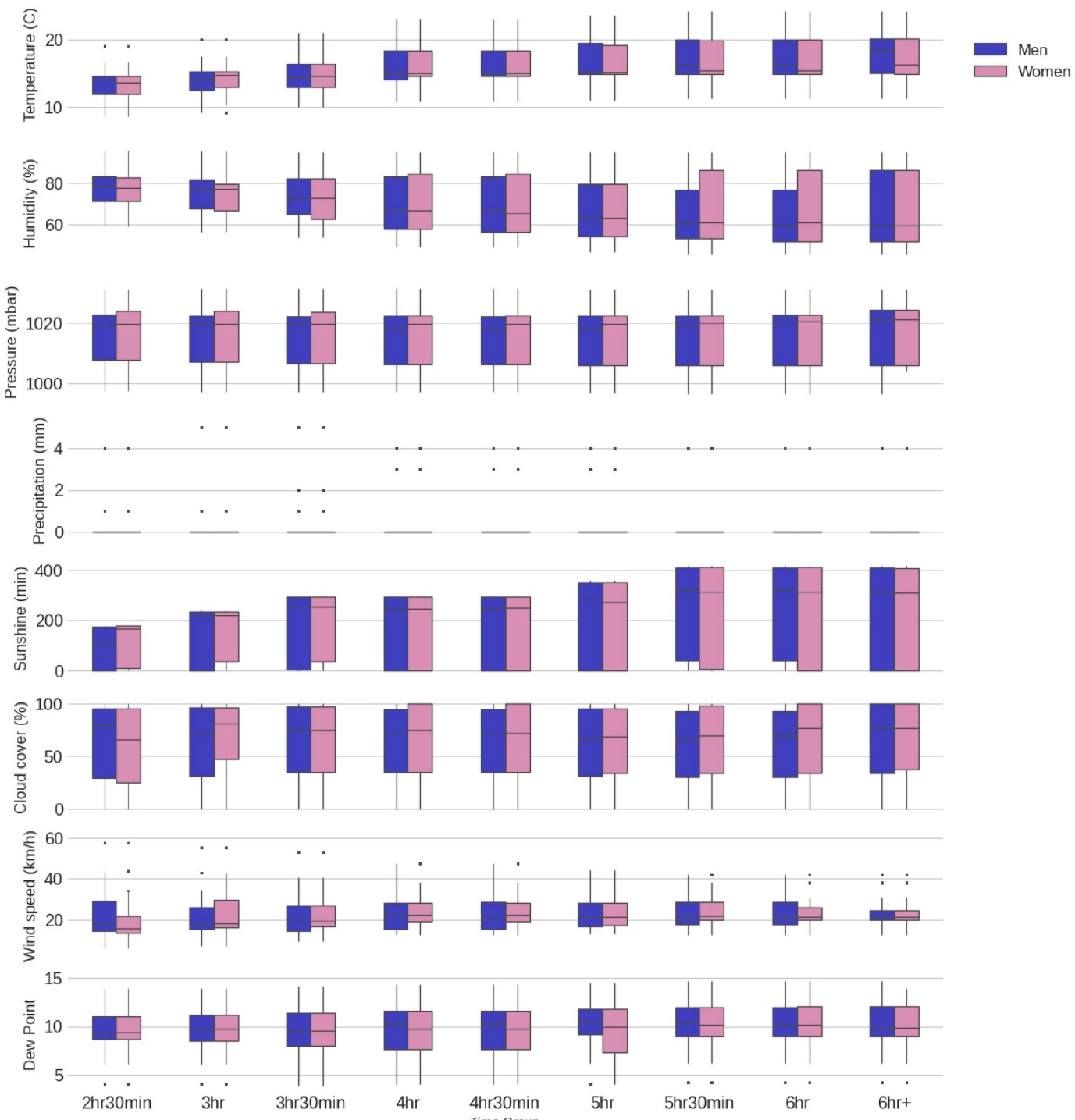

**Fig 4. Boxplot of weather conditions by finish time group and gender in Berlin Marathon between 1999 and 2019.**

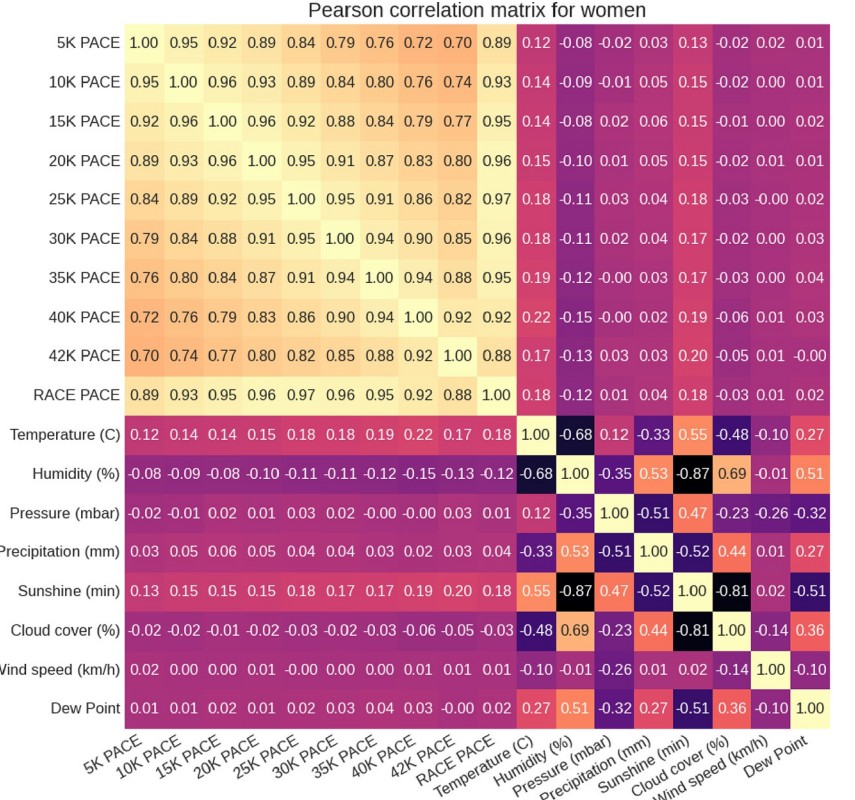

Pearson correlation matrix for women

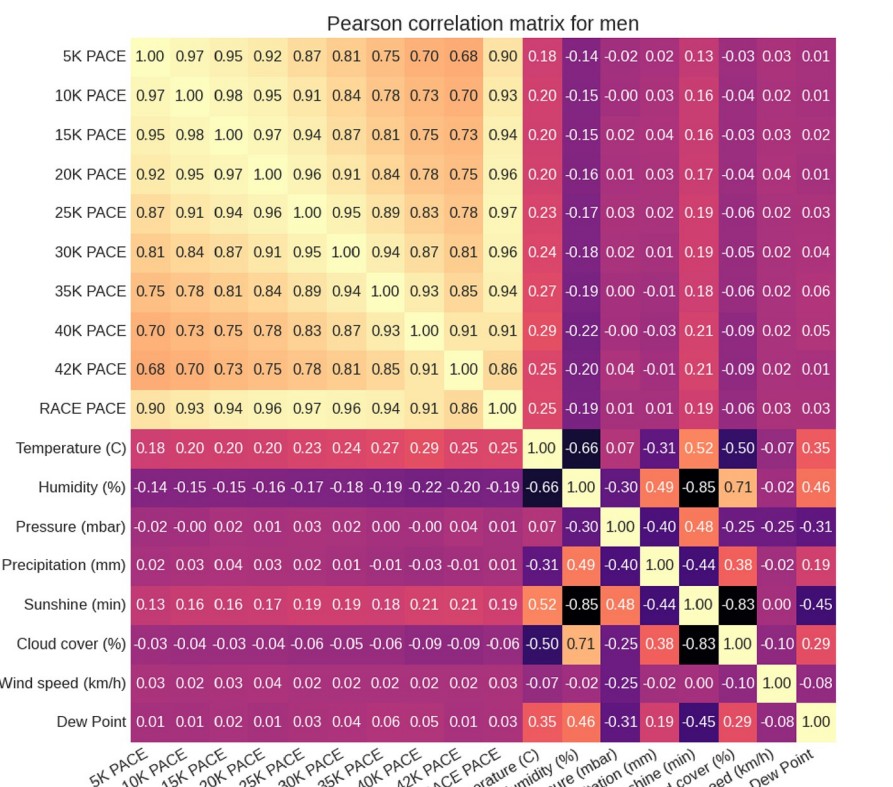

Pearson correlation matrix for men

**Fig 5. Pearson correlation matrixes of partial average pace and the full race vs. adjusted average weather values.** The values in the boxes are correlation coefficients (r-values).

When predicting power climbs, we added the 5 km speed column as a predictor (see **Table 3**; R-squared = 0.78), showing that we can predict the average race speed with much more accuracy by adding the first split distance average speed (Model 2). However, the evolution of the OLS regression models shows no significant predictive effect of weather conditions on pacing.

Recognizing the limitations of an MLR linear model, we processed the data using a more sophisticated tree-ensemble gradient-boosting algorithm. The XG Boost Regression model predicted the race pace using eight weather factors (temperature, barometric pressure, humidity, precipitation, sunshine, cloud cover, wind speed, and dew point). The highest accuracy was obtained with 90 estimators set as the optimal value for the final XG Boost model with an MAE of 0.28 min/km and an $R^2$ of 0.88. The high prediction rate was the result of the data used to train the model, where the imputed weather factors contained some information about the finishing time itself. With this data, the XG Boost model could evaluate the importance of the different weather factors, which we represented in **Fig 6** as SHAP aggregated values. The chart shows the different weather factors sorted by order of importance, where temperature shows the largest amount of variation.

## Discussion

The primary objective of this retrospective study was to examine whether various environmental factors, including temperature, barometric pressure, humidity, precipitation, sunshine,

**Table 2. OLS regression results for weather factors (Model 1).**

| | | | | | | | |
|---|---|---|---|---|---|---|---|
| | | | **OLS Regression Results** | | | | |
| 0.169 | R-squared: | | | | | y | Dep. Variable: |
| 0.169 | Adj. R-squared: | | | | | OLS | Model: |
| 1.694e+04 | F-statistic: | | | | | Least Squares | Method: |
| 0 | Prob (F-statistic): | | | | | Sun, 12 Mar 2023 | Date: |
| -8.9903e+05 | Log-Likelihood: | | | | | 19:46:49 | Time: |
| 1.798e+06 | AIC: | | | | | 668509 | No. Observations: |
| 1.798e+06 | BIC: | | | | | 668500 | Df Residuals: |
| | | | | | | 8 | Df Model: |
| | | | | | | nonrobust | Covariance Type: |
| 0.975] | [0.025 | P>\|t\| | t | std err | Coef | | |
| -22.241 | -22.955 | 0 | -124.218 | 0.182 | -22.5980 | | Const |
| 0.821 | 0.807 | 0 | 229.126 | 0.004 | 0.8142 | | Temperature (˚C) |
| 0.152 | 0.148 | 0 | 174.360 | 0.001 | 0.1501 | | Humidity (%) |
| 0.014 | 0.013 | 0 | 73.739 | 0 | 0.0133 | | Pressure (mbar) |
| -0.141 | -0.170 | 0 | -21.147 | 0.007 | -0.1558 | | Precipitation (mm) |
| -0.017 | -0.018 | 0 | -134.824 | 0 | -0.05714 | | Sunshine (min) |
| 0 | 0 | 0.666 | 0.431 | 6.73e-05 | 2.901e-05 | | Cloud cover (%) |
| 0.016 | 0.015 | 0 | 108.201 | 0 | 0.0156 | | Wind speed (km/h) |
| -0.823 | -0.838 | 0 | -213.244 | 0.004 | -0.8308 | | Dew point (˚C) |
| 0.001 | Durbin-Watson: | | | | 27263.434 | | Omnibus: |
| 30721.390 | Jarque-Bera (JB): | | | | 0 | | Prob(Omnibus): |
| 0 | Prob(JB): | | | | 0.515 | | Skew: |
| 1.64e+05 | Cond. No. | | | | 3.206 | | Kurtosis: |

**Table 3. OLS regression results for weather factors and 5 km running speeds (Model 2).**

| | | | | | | |
|---|---|---|---|---|---|---|
| | | | OLS Regression Results | | | |
| 0.787 | R-squared: | | | | y | Dep. Variable: |
| 0.787 | Adj. R-squared: | | | | OLS | Model: |
| 2.750e+05 | F-statistic: | | | | Least Squares | Method: |
| 0 | Prob (F-statistic): | | | | Sun, 12 Mar 2023 | Date: |
| -4.4334e+05 | Log-Likelihood: | | | | 19:46:49 | Time: |
| 8.867e+05 | AIC: | | | | 668509 | No. Observations: |
| 8.868e+05 | BIC: | | | | 668499 | Df Residuals: |
| | | | | | 9 | Df Model: |
| | | | | | nonrobust | Covariance Type: |
| 0.975] | [0.025 | P>|t| | t | std err | coef | |
| 2.788 | 2.420 | 0 | 27.769 | 0.094 | 2.6039 | const |
| 0.227 | 0.220 | 0 | 121.174 | 0.002 | 0.2238 | Temperature (˚C) |
| 0.044 | 0.042 | 0 | 96.964 | 0 | 0.0429 | Humidity (%) |
| 0.004 | 0.004 | 0 | 46.355 | 9.14e-05 | 0.0042 | Pressure (mbar) |
| -0.033 | -0.048 | 0 | -10.827 | 0.004 | -0.0404 | Precipitation (mm) |
| -0.002 | -0.002 | 0 | -26.816 | 6.62e-05 | -0.0118 | Sunshine (min) |
| 0.001 | 0.001 | 0 | 20.787 | 3.41e-05 | 0.0007 | Cloud cover (%) |
| 0.004 | 0.004 | 0 | 54.519 | 7.34e-05 | 0.0040 | Wind speed (km/h) |
| -0.210 | -0.218 | 0 | -105.997 | 0.002 | -0.2141 | Dew point (˚C) |
| -0.500 | -0.502 | 0 | -1394.522 | 0 | -0.5008 | 5K SPEED |
| 0.853 | Durbin-Watson: | | | | 128647.203 | Omnibus: |
| 276812.504 | Jarque-Bera (JB): | | | | 0 | Prob(Omnibus): |
| 0 | Prob(JB): | | | | 1.127 | Skew: |
| 1.67e+05 | Cond. No. | | | | 5.203 | Kurtosis: |

cloud cover, wind speed, and dew point, could impact the pacing of marathon runners participating in the 'Berlin Marathon'. We proposed three hypotheses: (*i*) an increase in temperature on race day would lead to a decrease in running speed, (*ii*) the effect of temperature on pacing would differ between men and women, and (*iii*) weather conditions could serve as a significant predictor of pacing. The key findings of the study were as follows: (1) running speed showed a significant negative correlation with temperature and a significant positive correlation with humidity, (2) this correlation was more pronounced among men than women, (3) a connection was observed between weather conditions and running speeds, it did not emerge as a significant predictor of pacing, and (4) there was a noticeable escalation in the strength of the relationship between temperature and race time for each successive split, potentially indicating heightened sensitivity to temperature and humidity as the race unfolded. Overall, average pace for the full race showed positive correlations with air temperature and minutes of sunshine for both men and women where these two factors slowed down runners.

## Impact of temperature on marathon performance

The well-established literature highlights a link between higher temperatures and declining performance among marathon runners [13, 27], with a pronounced effect in slower runners [8, 11]. According to thermodynamic principles, the body's heat production correlates with its mass, while heat dissipation through processes like radiation, convection, and evaporation is influenced by its surface area [28]. As temperatures rise, the efficiency of heat dissipation through radiation, convection, and conduction diminishes, prompting the body to rely more

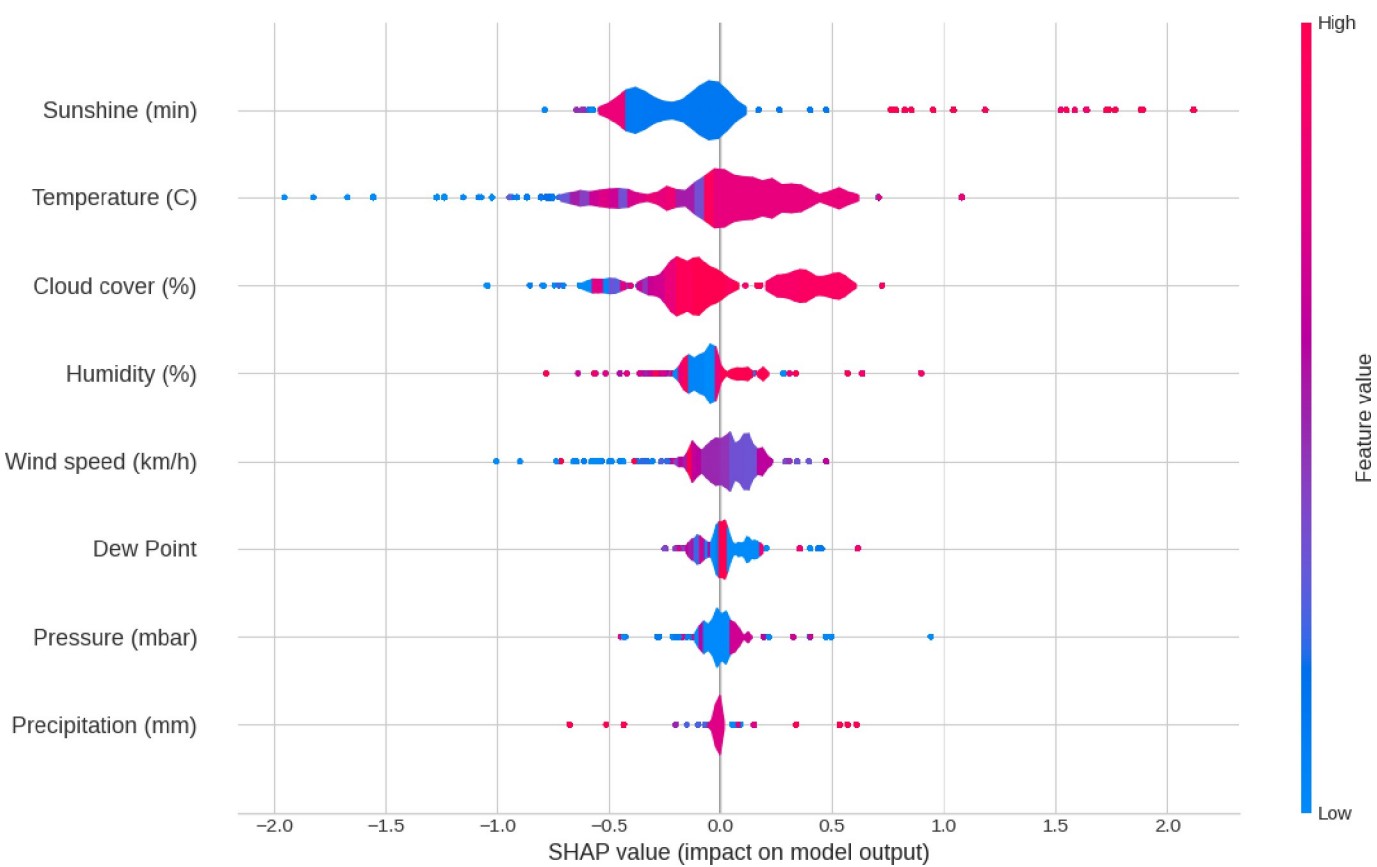

**Fig 6. SHAP aggregated values for each weather factor.**

on evaporative cooling. A study by Marino et al. revealed that runners with lower body mass generate and retain less heat than their heavier counterparts, a physiological characteristic that can impact performance even when maintaining similar running speeds [29]. However, it's crucial to consider that reduced running speed might also stem from fatigue and depletion of myocellular energy stores [30].

Another plausible explanation for the more pronounced decline in performance among slower runners stems from the timing of marathon races, typically commencing early in the morning. Faster runners tend to benefit from cooler morning temperatures, while slower runners contend with warmer conditions as the day progresses [15], largely influenced by the duration of sunlight exposure [18]. Previous research has demonstrated a gradual reduction in marathon race speed with rising temperatures [14], and it has been established that overcast skies and limited sunlight do not correlate with faster marathon race times [5]. To enhance the experience for slower marathoners, it could be preferable for races to conclude before 09:00 a. m. or commence after 07:00 p.m. This scheduling approach would offer athletes progressively ameliorating weather circumstances characterized by diminished sun exposure, gradual temperature increases or declines (morning or evening), and minimal humidity increase [31].

While the negative correlation between running speed and temperature might appear intuitive, the observed outcome was shaped by the circumstance that faster runners (with higher running speeds) participated in the race during the early morning hours, around 09:00 a.m., when temperatures were still relatively cool. In contrast, runners with slower paces (completing the race in over 4 hours) faced warmer temperatures during the afternoon. This

relationship has been accentuated by using adjusted averages in the dataset. The swifter runners encountered lower temperatures and elevated humidity levels, completing the course in under three hours. In contrast, leisure runners progressed at a slower pace, finishing the race later in the afternoon when temperatures were higher and humidity was lower.

### Gender differences in pacing and performance

The effects of temperature on pacing were less pronounced in women than in men. This could be attributed to the more selective roster of women participating. The lower number of women compared to male counterparts highlights a natural selection bias, where the average participating woman is more experienced and dedicated more time to training, resulting in evener pacing and, subsequently, less loss of speed [32]. With the increasing popularity of marathons and an increasing number of women, this selection bias should disappear in the future [33].

### Humidity and its effects on performance

Another significant discovery was the decline in humidity throughout the race, with a notable and positive correlation between running speed and humidity. This implies that as humidity diminished over the day, runners experienced a decrease in pace. The positive connection between running speed and humidity may find its rationale in the robust inverse correlation between temperature and relative humidity.

Humidity refers to the volume of water vapor present in the atmosphere. Elevated water vapor levels lead to higher humidity (www.weather.gov/lmk/humidity). The humidity level is also influenced by temperature. As temperatures rise, relative humidity decreases, whereas lower temperatures increase relative humidity (https://sciencing.com/temperature-ampamp-humidity-related-7245642.html). Within the context of physical activity in warm conditions, evaporation serves as a principal method for dissipating heat [34]. Humidity significantly affects heat dissipation through evaporation. Reduced relative humidity enhances sweating efficiency [35]. A lower presence of water vapor in the air allows sweat secretion to cool the body more effectively [35].

Prior research has indicated that heightened humidity levels can adversely impact athletic performance. Notably, moderate-intensity exercise in warm settings has shown a gradual decline in performance as relative humidity increases [36]. In controlled laboratory settings, investigations have highlighted reduced maximal aerobic capacity under elevated temperatures and varying humidity conditions [37]. The impact of humidity on marathon performance has also been explored in previous studies [9, 12]. Our findings confirm that a decrease in humidity was associated with a decrease in running performance. A comprehensive study examining the influence of weather conditions on 1,280,557 age group runners participating in the 'New York City Marathon' between 1970 and 2019 showed that higher temperatures, lower humidity, and lower wind speed corresponded with decreased performance. Specifically, the detriments of elevated humidity were more pronounced among men aged 40–59 and women aged 25–65 [9]. Another study focusing on the effects of weather on marathon running performance in the 'Stockholm Marathon' showed a significant and negative correlation between relative air humidity and the anomaly in marathon finishing times [38]. However, the authors attributed these effects of relative humidity to its negative correlation with air temperature [38]. Pragmatically, the rise in temperature and the race's duration likely held more explanatory power in elucidating performance decline than the decrease in humidity.

## Impact of other environmental factors

Given the absence of rainfall across all editions of the 'Berlin Marathon,' we were precluded from examining potential correlations between rain and running speed. It is known that rain can hinder marathon performance [38]. For instance, in the 'Stockholm Marathon,' there was a noteworthy and adverse correlation between rain and the anomaly in marathon finishing times [38]. Interestingly, contrasting these findings, the 'Boston Marathon' witnessed record times accomplished under the influence of drizzle [12].

The wind speed displayed a slight negative correlation with humidity, pressure, and dew point while demonstrating a weak positive correlation with the time of day. A study of master runners participating in the 'New York City Marathon' revealed a notable advantage associated with elevated wind speeds, particularly among men [9]. A study exploring the effects of wind assistance and resistance on a runner's forward motion recorded a net negative effect of winds exceeding the velocity of a runner on an oval track [39]. The extra energy expended against the wind will not be compensated by an equal gain when running with the wind [39]. As observed in the 'Berlin Marathon', our models have shown that wind speed doesn't lend itself as a detrimental predictor of the race pace. This could result from the urban landscape and mostly lower wind speeds throughout all marathon editions.

Cloud cover did not serve as a relevant predictor for performance. Recent findings have called into question the perceived importance of cloud cover for marathon running performance. Contrary to popular belief, neither cloud cover nor low sunshine are associated with faster marathon times. Data indicate that world records and the fastest marathon performances occur irrespective of cloud cover, and optimal temperatures for fast running are typically 10–15˚C [40].

Although we collected weather data at hourly intervals and initiated our analysis at 09:00 a.m. utilizing time-adjusted averages based on performance groups, it is important to note that the race itself featured interval starts spanning from 09:15 a.m. to 10:30 a.m. This variation depended on individual best times, often associated with half or marathon times, that each runner indicated during registration. Notably, novice marathon participants commenced the race at 10:30 a.m. within the final starting block (Block H). These differences could potentially have impacted the outcomes of our analyses. Furthermore, the absence of accessible data concerning anthropometric attributes, such as body height and mass, prevented us from investigating the interplay between environmental factors, performance, and athletes' physical characteristics. Additionally, we lacked information regarding these runners' experience and training volume [41].

## Limitations

A limitation of the study is that the correlation between running performance and temperature/humidity may be affected by the fact that slower runners tend to run for longer periods, particularly during later start times when the day is heating up and humidity levels are dropping. It is important to exercise caution when drawing causal relationships from this correlational data. As environmental temperature increases, relative humidity decreases but the total amount of water vapor in the air may not have changed (absolute humidity).

Relative humidity decreased with increasing marathon race times. This is likely a result of increasing temperatures during the day given the early start times and warming temperature as the day progresses and the longer the slower runners are exposed to this environmental process which is likely more of a correlational result. It is absolute humidity that determines sweat, evaporation and heat stress risk [42–44]. Furthermore, the course is not entirely flat and

the analyses did not include changes in elevation as a potential influence on pacing of the athletes.

## Conclusions

In summary, the increase in temperature and the decrease in humidity over the day exhibited an adverse association with the running speed of marathon runners competing in the 'Berlin Marathon'. This impact was more pronounced among men than women. Average pace for the full race showed positive correlations with the temperature and the minutes of sunshine for both men and women where these two factors slowed down runners. Another noteworthy observation was the increased sensitivity to temperature and humidity as the race unfolded. While weather variables did impact running speed and outcomes, their predictive significance on pacing remained limited. This underscores the importance of factoring in elements like training, body composition, and nutrition as more robust predictors of performance among age group runners. Future studies can investigate the influence of high humidity and precipitation on a marathon run in a tropical climate. Absolute humidity and changes in elevations should also be considered.

## Author Contributions

**Conceptualization:** Katja Weiss, Beat Knechtle.

**Data curation:** Elias Villiger.

**Formal analysis:** David Valero.

**Supervision:** Beat Knechtle.

**Writing – original draft:** Katja Weiss.

**Writing – review & editing:** Volker Scheer, Mabliny Thuany, Felipe J. Aidar, Raphael Fabrício de Souza, Ivan Cuk, Pantelis T. Nikolaidis, Thomas Rosemann, Beat Knechtle.

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
