## [Decision Letter · Decision Letter 0]

9 Jun 2024

PONE-D-24-15986Associations Between Environmental Factors and Running Performance in Age Group Runners: An Observational Study of the Berlin MarathonPLOS ONE

Dear Dr. Knechtle,

Thank you for submitting your manuscript to PLOS ONE. After careful consideration, we feel that it has merit but does not fully meet PLOS ONE’s publication criteria as it currently stands. Therefore, we invite you to submit a revised version of the manuscript that addresses the points raised during the review process.

Please note the followings when you revise the manuscript. The manuscript is required to improve writing. Specifically, the manuscript does not have a central focus that you would like to present to the reader. In the discission section, main hypotheses and findings are presented in the first paragraph but not fully discussed or compared to the previous literature. Also, there is limited physiology used to explain the results or why you were investigated to begin with. The manuscript may be of benefit to the literature but in its current form it is hard to fully grasp the importance of the data.

We look forward to receiving your revised manuscript.

Kind regards,

Hidenori Otani, Ph.D.

Academic Editor

PLOS ONE

Journal Requirements:

3. In the online submission form, you indicated that the datasets used and analyzed during the current study are available from the corresponding author on reasonable request. Weather data were from the 'Deutscher Wetterdienst' website (https://opendata.dwd.de/climate_environment/CDC/observations_germany/climate/hourly/) specifically for the 'Alexanderplatz' location, which is near the race venue. The athlete information was directly sourced from the official 'Berlin Marathon' website (www.bmw-berlin-marathon.com/). We acquired the complete dataset encompassing all races from 1999 to 2019 in JSON format, which was subsequently transformed into Excel format utilizing a tailored Python script.

Reviewers' comments:

Reviewer's Responses to Questions

**Comments to the Author**

1. Is the manuscript technically sound, and do the data support the conclusions?

Reviewer #1: Yes

Reviewer #2: Yes

2. Has the statistical analysis been performed appropriately and rigorously? 

Reviewer #1: Yes

Reviewer #2: Yes

3. Have the authors made all data underlying the findings in their manuscript fully available?

Reviewer #1: Yes

Reviewer #2: Yes

4. Is the manuscript presented in an intelligible fashion and written in standard English?

Reviewer #1: Yes

Reviewer #2: Yes

5. Review Comments to the Author

Reviewer #1: The manuscript titled “Associations Between Environmental Factors and Running Performance in Age Group Runners: An Observational Study of the Berlin Marathon” found that increasing environmental temperatures negatively impacted the overall running speed. The examination is unique due to the large data set (520, 521 men, 147,988 women), the multiple environmental parameters, and extensive statistical analyses. The analyses and discussion, while interesting, were difficult to wade through. Specifically, there was some disconnect between the main hypotheses, analysis presentation, and discussion of these findings. I have provided comments in the hope that it will streamline the main findings and bring the importance of these analysis to the forefront.

Comments

Title: The title does not reflect the analyses. The title includes “age group” runners, but no analysis on age was completed. Please reword.

Abstract: Please reconsider your use of the term “Recreational Runner”. Runners completing the marathon in 2:30 would still display a V02 in the 90% for the population, especially is they were female, and would have likely taken months of regimented training to accomplish this time. The conclusion that these results could be used to “manage hydration and running tempo” was not addressed anywhere in the discussion. Additionally, your analysis was a retrospective analysis on how running speed may have changed with environment and there was no predictive effect on pacing. This fact should be considered when presenting the results in the discussion as well.

Introduction: Please double check references. Many references used within the paper apply to multiple statements but are excluded, specifically around past examinations of marathon performance and environmental conditions. A reference which may assist in the analysis of cloud cover and solar load (PMID 17986912). The physiology presented in lines 102-112 only address part of heat production and does not appropriately address heat dissipation and why the environment would be of importance. Please see and reference PMID 13930723. Line 114-121: the majority of the introduction is about environmental temperature, but you mention that you want to investigate other environmental factors such as barometric pressure, please provide greater rational for investigating these other factors.

Methods: Please identify if you are using Relative or Absolute humidity. There is a correlation between absolute humidity and heat dissipation, less so for relative humidity. Did you account for improved performance with time. As you mention in line 89, Berlin has had the most world records. Clearly there is a shift to increasing times for all athletes for the past 24 years. Did you account for course topography when calculating pacing and comparing between each 5K? Did you obtain humidity from the Deutscher Wetterdienst? It is not listed in lines 148-151. Did you average the hourly weather parameters for the length of your time grouping? If so, how did you account for hourly weather data, yet had time groups of 30 minutes?

Results: There are extensive results and figures that do not address the main hypotheses presented by the authors. For example, the average running speed of all athletes was 10.5 km/hr and figure 1 shows average running speed by all runners per year, per 5K but weather is not incorporated, nor is there error or variability presented. There is so much data presented in multiple ways, it is unknown what the authors would like the reader to focus on, and/or what data directly address the authors stated hypotheses. Were men’s and women’s changes in pacing directly compared? These seem to be separate analyses.

Discussion: The discussion starts with the main findings that address the author’s hypothesis but then the rest of the discussion is broken up into sections discussing each environmental factor. It would help the reader to have sub-headings that specifically address each hypothesis, especially since many environmental facts had no effect on runner pacing.

Reviewer #2: This is an interesting study examining historical finishing times at the Berlin Marathon and associations with weather and environmental factors. The authors should be commended on their handling of such a large data set.

I have a few comments that may be helpful.

Title: Use of ‘age-grade’. Why this term specifically? Is this a common term in the running community? The inclusion for the participants needs further clarification in the methods section. If elites weren’t used what was the cut-off for this? If age-grade is the term, the participants are then not split by age.

L110 – Is the greater decline in faster runners a result of trying to compete for the win and therefore possibly not following their pacing strategy?

L140 – methods – subjects – see earlier comment. I think more is needed in this section. Also presumably some runners participated in multiple years. Did you consider analysing this data as a subset to compare the effect of environmental conditions? I appreciate there will be lots of limitations with this but it may be a consideration.

L149 – Is there a way to measure sunshine in intensity i.e W/m2?

L230 – Were these variables measured to the degree of accuracy suggested by the number of decimal places?

Figure 4 – Is it possible to have a minimum, maximum and average value for precipitation? Was it noy just one value?

Fig 6 – For sunshine, would you noy expect to see higher exposure with slower runners if measured in minutes? The slower runners would have been exposed to all of the sun for the faster runners and additional sun?

Fig 7 – I appreciate different colours for males and females but using the same colour scheme may help to visually compare the strength of each relationship.

Discussion – Is there a difference in sweat rates between faster and slower runners? I.e greater sweat rates/ more efficient. Plus increased metabolic heat production?

L429-438 – I wonder how relevant this is to the study?

L459 – Is this line correct? Does it contrast with L440-441?

6. PLOS authors have the option to publish the peer review history of their article (what does this mean?). If published, this will include your full peer review and any attached files.

Reviewer #1: No

Reviewer #2: No

---

## [Author Response · Author response to Decision Letter 0]

15 Jul 2024

Editor:

Please note the followings when you revise the manuscript. The manuscript is required to improve writing. Specifically, the manuscript does not have a central focus that you would like to present to the reader. In the discission section, main hypotheses and findings are presented in the first paragraph but not fully discussed or compared to the previous literature. Also, there is limited physiology used to explain the results or why you were investigated to begin with. The manuscript may be of benefit to the literature but in its current form it is hard to fully grasp the importance of the data.

Answer: We thank you for the thorough review and have made an effort to address all stated issues. We have reworked our discussion section and curtailed our figures to present a more focused and readable manuscript while highlighting the important results of our study.

Reviewer #1: 

The manuscript titled “Associations Between Environmental Factors and Running Performance in Age Group Runners: An Observational Study of the Berlin Marathon” found that increasing environmental temperatures negatively impacted the overall running speed. The examination is unique due to the large data set (520, 521 men, 147,988 women), the multiple environmental parameters, and extensive statistical analyses. The analyses and discussion, while interesting, were difficult to wade through. Specifically, there was some disconnect between the main hypotheses, analysis presentation, and discussion of these findings. I have provided comments in the hope that it will streamline the main findings and bring the importance of these analysis to the forefront.

Answer: We agree with the expert reviewer and have addressed these concerns as best as we could.

Comments

Title: The title does not reflect the analyses. The title includes “age group” runners, but no analysis on age was completed. Please reword.

Answer: We agree with the expert reviewer and have amended the title and removed the term age group.

Abstract: Please reconsider your use of the term “Recreational Runner”. Runners completing the marathon in 2:30 would still display a V02 in the 90% for the population, especially is they were female, and would have likely taken months of regimented training to accomplish this time. 

Answer: We have changed the term to amateur runners to better classify these participants who do not receive monetary compensation for participating. We agree that the outliers who fall into the under 2:30 group are exceptional runners who have prepared for these events for months.

The conclusion that these results could be used to “manage hydration and running tempo” was not addressed anywhere in the discussion. 

Answer: We agree with the expert reviewer and have aligned the conclusion statement in the abstract with the one in the conclusion section.

Additionally, your analysis was a retrospective analysis on how running speed may have changed with environment and there was no predictive effect on pacing. This fact should be considered when presenting the results in the discussion as well.

Answer: We agree with the expert reviewer and have highlighted this as suggested.

Introduction: Please double check references. Many references used within the paper apply to multiple statements but are excluded, specifically around past examinations of marathon performance and environmental conditions. A reference which may assist in the analysis of cloud cover and solar load (PMID 17986912). The physiology presented in lines 102-112 only address part of heat production and does not appropriately address heat dissipation and why the environment would be of importance. Please see and reference PMID 13930723. Answer: Answer: We agree with the expert reviewer and have included the suggested references in the manuscript.

Line 114-121: the majority of the introduction is about environmental temperature, but you mention that you want to investigate other environmental factors such as barometric pressure, please provide greater rational for investigating these other factors.

Answer: We agree with the expert reviewer and have added a greater rationale to our Introduction to address this shortcoming.

Methods: Please identify if you are using Relative or Absolute humidity. There is a correlation between absolute humidity and heat dissipation, less so for relative humidity. Did you account for improved performance with time. 

Answer: We have used relative humidity in our analysis, as highlighted by the use of percents as a unit instead of g/m3. As for the improved performance with time, we could not account for the participant’s acclimatization because we don’t have the data from which we could derive it.

As you mention in line 89, Berlin has had the most world records. Clearly there is a shift to increasing times for all athletes for the past 24 years.

Answer: Indeed, the pacing strategies of the best marathon runners in the world have changed over the last 50 years, as stated in the referenced paper doi: 10.1080/17461391.2018.1450899.

 Did you account for course topography when calculating pacing and comparing between each 5K? 

Answer: We did not account for the course topography since the Berlin Marathon features a flat course.

Did you obtain humidity from the Deutscher Wetterdienst? It is not listed in lines 148-151. Answer: Yes. We have forgotten to add it. Thank you for the hint. We have amended our manuscript.

Did you average the hourly weather parameters for the length of your time grouping? If so, how did you account for hourly weather data, yet had time groups of 30 minutes?

Answer: We have classified all race records in 30-minute groups, and then the average values of the weather variables for each group were calculated according to the following scheme:

It seems that we made an error in our calculation. For the sunshine (minutes) and precipitation (mm) variables, rather than averages we should have calculated cumulative values over the race time. We have corrected our calculation accordingly.

Results: There are extensive results and figures that do not address the main hypotheses presented by the authors. 

Answer: We agree with the expert reviewer and have removed some of the unnecessary figures.

For example, the average running speed of all athletes was 10.5 km/hr and figure 1 shows average running speed by all runners per year, per 5K but weather is not incorporated, nor is there error or variability presented. 

Answer: We have tried to improve the readability of these charts by removing the shading and using a more colorblind scheme including dashed, dotted, and full lines. We also added the series for the final spurt (called “42K PACE”) which shows some interesting observations. 

There is so much data presented in multiple ways, it is unknown what the authors would like the reader to focus on, and/or what data directly address the authors stated hypotheses. Were men’s and women’s changes in pacing directly compared? These seem to be separate analyses.

Answer: We agree with the expert reviewer and have removed some of the unnecessary figures and cleaned up the rest.

Discussion: The discussion starts with the main findings that address the author’s hypothesis but then the rest of the discussion is broken up into sections discussing each environmental factor. It would help the reader to have sub-headings that specifically address each hypothesis, especially since many environmental facts had no effect on runner pacing.

Answer: We agree with the expert reviewer and have added the relevant subheading to the manuscript.

Reviewer #2: 

This is an interesting study examining historical finishing times at the Berlin Marathon and associations with weather and environmental factors. The authors should be commended on their handling of such a large data set.

I have a few comments that may be helpful.

Title: Use of ‘age-grade’. Why this term specifically? Is this a common term in the running community? The inclusion for the participants needs further clarification in the methods section. If elites weren’t used what was the cut-off for this? If age-grade is the term, the participants are then not split by age.

Answer: Thank you for pointing out our error. We have removed the term from the manuscript. Regarding the cut-off point, we have included all records to get the most generalized results possible.

L110 – Is the greater decline in faster runners a result of trying to compete for the win and therefore possibly not following their pacing strategy?

Answer: Indeed, the mismanaging of the set pacing strategy would result in a greater decline. However, it is unknown what percentage of runners have been impacted by their own ambition and how many of them just didn’t have a planned pacing strategy. It could, however, be a topic of a future study.

L140 – methods – subjects – see earlier comment. I think more is needed in this section. Also presumably some runners participated in multiple years. Did you consider analysing this data as a subset to compare the effect of environmental conditions? I appreciate there will be lots of limitations with this but it may be a consideration.

Answer: We regrettably did not take into consideration the fact that some runners may have taken part in different editions of the race. Given the large sample size and the fact the race records were registered throughout a 20-year period, we did not think it would be too relevant.

L149 – Is there a way to measure sunshine in intensity i.e W/m2?

Answer: Regrettably, we did not have data related to sunshine intensity. The weather stations data of the provided 'Deutscher Wetterdienst' only the sunshine duration (https://opendata.dwd.de/climate_environment/CDC/observations_germany/climate/hourly/sun/DESCRIPTION_obsgermany_climate_hourly_sun_en.pdf).

L230 – Were these variables measured to the degree of accuracy suggested by the number of decimal places?

Answer: No. The original, hourly values of the weather variables are provided as integers. The decimal places only appear as a result of the calculations of mean, std and the other statistical values. 

Figure 4 – Is it possible to have a minimum, maximum and average value for precipitation? Was it noy just one value?

Answer: It is possible, given the weather data is provided hourly (so, for instance, on a given day, we could experience 0 mm of precipitation in the first hour, 1 mm in the second hour, and 2 mm in the third hour, with no further rain during the remaining hours. So we would have min=0, average=3/8, and max=2 in an hour. Nonetheless, we have decided to remove this chart, as the sunshine and precipitation variables are not aligned with the new cumulative values used in the boxplots.

Fig 6 – For sunshine, would you noy expect to see higher exposure with slower runners if measured in minutes? The slower runners would have been exposed to all of the sun for the faster runners and additional sun?

Answer: Thank you for pointing this out. We have fixed the issue and we can indeed see from the boxplots that slower runners get longer sun exposition. As a result of the preceding changes this is now Figure 5.

Fig 7 – I appreciate different colours for males and females but using the same colour scheme may help to visually compare the strength of each relationship.

Answer: Thank you for the hint. We have changed it to the same color scheme. As a result of the preceding changes, this is now Figure 6.

Discussion – Is there a difference in sweat rates between faster and slower runners? I.e greater sweat rates/ more efficient. Plus increased metabolic heat production?

Answer: Sweat rates are highly dependent on each individual physiology and we did not have any data from which we could deduce the sweat rates per runner.

L429-438 – I wonder how relevant this is to the study?

Answer: Thank you for pointing out our musings. We have removed the paragraph to compress the study to more relevant information.

L459 – Is this line correct? Does it contrast with L440-441?

Answer: That line should be correct by our understanding. Decreasing humidity was associated with a decrease in the pace and running performance.

---

## [Decision Letter · Decision Letter 1]

13 Aug 2024

PONE-D-24-15986R1Associations Between Environmental Factors and Running Performance: An Observational Study of the Berlin MarathonPLOS ONE

Dear Dr. Knechtle,

Thank you for submitting your manuscript to PLOS ONE. After careful consideration, we feel that it has merit but does not fully meet PLOS ONE’s publication criteria as it currently stands. Therefore, we invite you to submit a revised version of the manuscript that addresses the points raised during the review process.

We look forward to receiving your revised manuscript.

Kind regards,

Hidenori Otani, Ph.D.

Academic Editor

PLOS ONE

Reviewers' comments:

Reviewer's Responses to Questions

**Comments to the Author**

1. If the authors have adequately addressed your comments raised in a previous round of review and you feel that this manuscript is now acceptable for publication, you may indicate that here to bypass the “Comments to the Author” section, enter your conflict of interest statement in the “Confidential to Editor” section, and submit your "Accept" recommendation.

Reviewer #1: (No Response)

Reviewer #2: (No Response)

2. Is the manuscript technically sound, and do the data support the conclusions?

Reviewer #1: Yes

Reviewer #2: Yes

3. Has the statistical analysis been performed appropriately and rigorously? 

Reviewer #1: Yes

Reviewer #2: Yes

4. Have the authors made all data underlying the findings in their manuscript fully available?

Reviewer #1: Yes

Reviewer #2: Yes

5. Is the manuscript presented in an intelligible fashion and written in standard English?

Reviewer #1: Yes

Reviewer #2: Yes

6. Review Comments to the Author

Reviewer #1: Thank you for considering my suggestions but I do not feel they have all been properly addressed.

Thank you for revising the discussion. It is much easier to follow and shows how the current work builds upon the previous literature. Although, the introduction could use better refinement as well to set-up the purpose/better align with the hypotheses of the study.

Still my main concerns follow which have not been fully addressed.

1) As environmental temperature increases, relative humidity decreases but the total amount of water vapor in the air may not have changed (absolute humidity). You have shown that with increasing marathon times that relative humidity decreases. This is likely a result of increasing temperatures during the day given the early start times and warming temperature as the day progresses AND the longer the slower runners are exposed to this environmental process. So, this is likely more of a correlational result. It is absolute humidity that determines sweat, evaporation and heat stress risk. Please see PMID: 37255302 and PMID: 29368184, PMID: 38695357. This should be acknowledged and properly addressed.

2) The authors conclude that coaches and athletes can use this information to manage their performance? This has not been addressed? How could this information be used. What the authors are presenting is a retrospective analysis of how environmental factors affected pace.

3) The authors did not consider course topography and replied that the course is “Flat”. While it may be flat in comparison to other marathons, it does climb ~20 m with a slight uphill from 20-25km and a bigger incline from 25-27km then a downhill from 27-32km. The authors state in line 208 that running speed “stayed almost constant through 15-20 km. at 20 km was where there is a slight uphill.

4) All figures do not directly address the hypotheses. For example: Figure 1 and 2 do not address how an environmental factor affects pace. Figure 3, fine. Figure 4. Comparison of pace between genders but does not include how temperature may differentially affect performance. Given that these individuals were binned by finishing time, what does this tell the reader? Figure 5, the first to incorporate environmental factors that each division of finishers is exposed to. Figure 6. Why not include overall finishing time in matrix, that is one of the conclusions.

5) Discussion: While better organized includes a section (line 366-371) that is not relevant to the findings and not 100% physiologically accurate. Additionally, metabolic rate is the driver for heat production (line 361).

Although not part of my original critique can the authors confirm that the results indicate that running speed (pace) decreases from 5km to 10, 15, 20, 25, 30, 35, 40km in both men and women and that increasing environmental temperatures increase this rate of slowing but more so in the men than women. This should be very explicitly stated as a result.

Reviewer #2: Thank you for the detailed responses. My comments have been addressed appropriately apart from one.

L230 – Were these variables measured to the degree of accuracy suggested by the number of

decimal places?

Answer: No. The original, hourly values of the weather variables are provided as integers.

The decimal places only appear as a result of the calculations of mean, std and the other

statistical values.

If the original values were provided as integers then standard practice would be to report them to that. Reporting to decimal places suggests that that level of accuracy and precision was there when it wasn't. Please amend in line with this.

7. PLOS authors have the option to publish the peer review history of their article (what does this mean?). If published, this will include your full peer review and any attached files.

Reviewer #1: No

Reviewer #2: **Yes: **Stephen Mears

---

## [Author Response · Author response to Decision Letter 1]

31 Aug 2024

Reviewer #1: Thank you for considering my suggestions but I do not feel they have all been properly addressed.

Thank you for revising the discussion. It is much easier to follow and shows how the current work builds upon the previous literature. Although, the introduction could use better refinement as well to set-up the purpose/better align with the hypotheses of the study.

Still my main concerns follow which have not been fully addressed.

We thank the expert reviewer for his/her comments and tried to improve again

1) As environmental temperature increases, relative humidity decreases but the total amount of water vapor in the air may not have changed (absolute humidity). You have shown that with increasing marathon times that relative humidity decreases. This is likely a result of increasing temperatures during the day given the early start times and warming temperature as the day progresses AND the longer the slower runners are exposed to this environmental process. So, this is likely more of a correlational result. It is absolute humidity that determines sweat, evaporation and heat stress risk. Please see PMID: 37255302 and PMID: 29368184, PMID: 38695357. This should be acknowledged and properly addressed.

Answer: We agree with the expert reviewer and added in the limitations ‘As environmental temperature increases, relative humidity decreases but the total amount of water vapor in the air may not have changed (absolute humidity). Relative humidity decreased with increasing marathon race times. This is likely a result of increasing temperatures during the day given the early start times and warming temperature as the day progresses and the longer the slower runners are exposed to this environmental process which is likely more of a correlational result. It is absolute humidity that determines sweat, evaporation and heat stress risk’. We added the suggested references of Prof. Ollie Jay. We also added in the conclusions ‘Absolute humidity and changes in elevations should also be considered.’.

2) The authors conclude that coaches and athletes can use this information to manage their performance? This has not been addressed? How could this information be used. What the authors are presenting is a retrospective analysis of how environmental factors affected pace.

Answer: We agree with the expert reviewer and delete this statement because other variables have a higher influence on pacing and weather cannot be changed on race day.

3) The authors did not consider course topography and replied that the course is “Flat”. While it may be flat in comparison to other marathons, it does climb ~20 m with a slight uphill from 20-25km and a bigger incline from 25-27km then a downhill from 27-32km. The authors state in line 208 that running speed “stayed almost constant through 15-20 km. at 20 km was where there is a slight uphill.

Answer: We add in the section ‘The race’ the details ‘In detail, it climbs ~20 m with a slight uphill from 20-25 km and a bigger incline from 25-27 km then a downhill from 27-32 km’ and insert in the limitations ‘The course is not entirely flat and the analyses did not include changes in elevation as a potential influence on pacing of the athletes’ to account for this aspect.

4) All figures do not directly address the hypotheses. For example: Figure 1 and 2 do not address how an environmental factor affects pace. Figure 3, fine. Figure 4. Comparison of pace between genders but does not include how temperature may differentially affect performance. Given that these individuals were binned by finishing time, what does this tell the reader? Figure 5, the first to incorporate environmental factors that each division of finishers is exposed to. Figure 6. Why not include overall finishing time in matrix, that is one of the conclusions.

Answer: Please see answers to comments below where we think we should have most of these figures in the manuscript. We only deleted figure 4. Figure 5 (former figure 6), the big correlation matrix, has been updated to include the average pace of the full race, as requested by the reviewer. The main observation is that the average pace for the full race shows positive correlations with the temperature and the minutes of sunshine for both men and women (so these two factors slow down runners).

We added the aspect that the average pace for the full race shows positive correlations with the temperature and the minutes of sunshine for both men and women also in the abstract and in the results section. However, the primary aim of the study was the influence on weather on pacing in the runners.

5) Discussion: While better organized includes a section (line 366-371) that is not relevant to the findings and not 100% physiologically accurate. Additionally, metabolic rate is the driver for heat production (line 361).

Answer: We agree with the expert reviewer and deleted that section since it has no direct link to performance and weather.

Although not part of my original critique can the authors confirm that the results indicate that running speed (pace) decreases from 5km to 10, 15, 20, 25, 30, 35, 40km in both men and women and that increasing environmental temperatures increase this rate of slowing but more so in the men than women. This should be very explicitly stated as a result.

Answer: We agree with the expert reviewer and pace decreases with the race distance (most likely as a result of fatigue), with the exception of the pace of the final spurt, where the athletes seem to make a final effort. This is what is shown in the following chart:

From the big correlation matrixes, the second part of the reviewer´s comment can also be deducted: for instance, the correlations between the pace at each partial split (5K, 10K, etc.) and the temperature are higher for men than for women (this means the variation of 1° Celsius causes a higher variation in the pace of men than that of women). This pattern is similar for the minutes of sunshine, although to a lesser extent.

Reviewer #2: Thank you for the detailed responses. My comments have been addressed appropriately apart from one.

L230 – Were these variables measured to the degree of accuracy suggested by the number of

decimal places?

Answer: No. The original, hourly values of the weather variables are provided as integers.

The decimal places only appear as a result of the calculations of mean, std and the other

statistical values.

If the original values were provided as integers then standard practice would be to report them to that. Reporting to decimal places suggests that that level of accuracy and precision was there when it wasn't. Please amend in line with this.

Answer: Unfortunately, we cannot agree with the reviewer here. When we calculate average (mean), standard deviation (std) and percentile values from a list of integers, the resulting values will be non-integer in most cases. There is nothing bad with it, and this makes no statement about the accuracy or resolution of the measurement equipment or its readings. Note also these decimals only appear in the calculated values, but not in the maximum and minimum values (as those are original integer readings – i.e. Min of 7.00 and Max of 27.00).

---

## [Decision Letter · Decision Letter 2]

18 Sep 2024

PONE-D-24-15986R2Associations Between Environmental Factors and Running Performance: An Observational Study of the Berlin MarathonPLOS ONE

Dear Dr. Knechtle,

Thank you for submitting your manuscript to PLOS ONE. After careful consideration, we feel that it has merit but does not fully meet PLOS ONE’s publication criteria as it currently stands. Therefore, we invite you to submit a revised version of the manuscript that addresses the points raised during the review process.

We look forward to receiving your revised manuscript.

Kind regards,

Hidenori Otani, Ph.D.

Academic Editor

PLOS ONE

Journal Requirements:

Reviewers' comments:

Reviewer's Responses to Questions

**Comments to the Author**

1. If the authors have adequately addressed your comments raised in a previous round of review and you feel that this manuscript is now acceptable for publication, you may indicate that here to bypass the “Comments to the Author” section, enter your conflict of interest statement in the “Confidential to Editor” section, and submit your "Accept" recommendation.

Reviewer #1: All comments have been addressed

Reviewer #2: (No Response)

2. Is the manuscript technically sound, and do the data support the conclusions?

Reviewer #1: Yes

Reviewer #2: Yes

3. Has the statistical analysis been performed appropriately and rigorously? 

Reviewer #1: Yes

Reviewer #2: Yes

4. Have the authors made all data underlying the findings in their manuscript fully available?

Reviewer #1: Yes

Reviewer #2: Yes

5. Is the manuscript presented in an intelligible fashion and written in standard English?

Reviewer #1: Yes

Reviewer #2: Yes

6. Review Comments to the Author

Reviewer #1: Thank you for considering my suggestions. I believe this manuscript will add nicely to the previous examinations of Marathon performance and weather analyses.

Reviewer #2: Thank you for the response but I respectively disagree. It is common scientific practice that you report mean/ SD values to the same level as originally measured by rounding the values for mean and standard deviation. If temperature was 15°C,16°C etc then the mean should be rounded to a whole integer. If it was 15.5°C, 16.7°C it should be rounded to 1 decimal place. Presumably the mean and SD output for all was 2 decimal places – some would have been more i.e. 3+ so the question is why it was not presented to that level of output. The number of decimal places showcases the accuracy of your measurement. For example, presenting rainfall to 2 decimal places suggest that it was measured to the nearest 0.01 mm which I doubt was possible.

7. PLOS authors have the option to publish the peer review history of their article (what does this mean?). If published, this will include your full peer review and any attached files.

Reviewer #1: No

Reviewer #2: No

---

## [Author Response · Author response to Decision Letter 2]

27 Sep 2024

Reviewer #1: Thank you for considering my suggestions. I believe this manuscript will add nicely to the previous examinations of Marathon performance and weather analyses.

Answer: We thank the expert reviewer for his/her comment, no further changes are required.

Reviewer #2: Thank you for the response but I respectively disagree. It is common scientific practice that you report mean/ SD values to the same level as originally measured by rounding the values for mean and standard deviation. If temperature was 15°C,16°C etc then the mean should be rounded to a whole integer. If it was 15.5°C, 16.7°C it should be rounded to 1 decimal place. Presumably the mean and SD output for all was 2 decimal places – some would have been more i.e. 3+ so the question is why it was not presented to that level of output. The number of decimal places showcases the accuracy of your measurement. For example, presenting rainfall to 2 decimal places suggest that it was measured to the nearest 0.01 mm which I doubt was possible.

Answer: I have adapted all numbers in Table 1 to the closest integer since the original weather data was provided with integer resolution. Also edited / added to this respect in the Methods / Weather data section.

---

## [Decision Letter · Decision Letter 3]

1 Oct 2024

Associations Between Environmental Factors and Running Performance: An Observational Study of the Berlin Marathon

PONE-D-24-15986R3

Dear Dr. Knechtle,

We’re pleased to inform you that your manuscript has been judged scientifically suitable for publication and will be formally accepted for publication once it meets all outstanding technical requirements.

Kind regards,

Hidenori Otani, Ph.D.

Academic Editor

PLOS ONE

Reviewers' comments:

Reviewer's Responses to Questions

**Comments to the Author**

1. If the authors have adequately addressed your comments raised in a previous round of review and you feel that this manuscript is now acceptable for publication, you may indicate that here to bypass the “Comments to the Author” section, enter your conflict of interest statement in the “Confidential to Editor” section, and submit your "Accept" recommendation.

Reviewer #2: All comments have been addressed

2. Is the manuscript technically sound, and do the data support the conclusions?

Reviewer #2: Yes

3. Has the statistical analysis been performed appropriately and rigorously? 

Reviewer #2: Yes

4. Have the authors made all data underlying the findings in their manuscript fully available?

Reviewer #2: Yes

5. Is the manuscript presented in an intelligible fashion and written in standard English?

Reviewer #2: Yes

6. Review Comments to the Author

Reviewer #2: Thank you for addressing the comment. This paper will be very interesting to the readers in this area.

7. PLOS authors have the option to publish the peer review history of their article (what does this mean?). If published, this will include your full peer review and any attached files.

Reviewer #2: No

---

## [Editor Report · Acceptance letter]

7 Oct 2024

PONE-D-24-15986R3 

PLOS ONE

Dear Dr. Knechtle, 

I'm pleased to inform you that your manuscript has been deemed suitable for publication in PLOS ONE. Congratulations! Your manuscript is now being handed over to our production team.

Kind regards, 

on behalf of

Dr. Hidenori Otani 

Academic Editor

PLOS ONE